# TRANSFORMERS SELF-ORGANIZE LIKE NEWBORN VISUAL SYSTEMS WHEN TRAINED IN PRENATAL WORLDS

## ABSTRACT

Do transformers learn like brains? A key challenge in addressing this question is that transformers and brains are trained on fundamentally different data. Brains are initially "trained" on prenatal sensory experiences (e.g., retinal waves), whereas transformers are typically trained on large datasets that are not biologically plausible. We reasoned that if transformers learn like brains, then they should develop the same structure as newborn brains when exposed to the same prenatal data. To test this prediction, we simulated prenatal visual input using a retinal wave generator. Then, using self-supervised temporal learning, we trained transformers to adapt to those retinal waves. During training, the transformers spontaneously developed the same structure as newborn visual systems: (1) early layers became sensitive to edges, (2) later layers became sensitive to shapes, and (3) the models developed larger receptive fields across layers. The organization of newborn visual systems emerges spontaneously when transformers adapt to a prenatal visual world. This developmental convergence suggests that brains and transformers learn in common ways and follow the same general fitting principles.

## 1 INTRODUCTION

A core goal in artificial intelligence (AI) is to build machines that learn like brains. But how will we know when we have succeeded? The behavior of all learning systems depends on both the learning algorithm and training data from which the system learns. Thus, to compare learning across brains and machines, we must give them the same training data. While straightforward in principle, this is challenging in practice.

Brains have a prenatal development phase, which provides a unique set of experiences that shape the initial organization of the brain. For example, retinal waves are widespread during prenatal development and play a direct causal role in the development of the visual system (Meister et al., 1991; Torborg & Feller, 2005; Ackman et al., 2012). In contrast, machine-learning systems do not have a prenatal developmental phase, meaning the initial training data for "initializing" brains versus machines is different. This makes it difficult—if not impossible—to accurately compare learning across brains versus machines, since differences in learning could be due to the algorithm, training data, or some combination of the two factors.

We performed a rigorous test of whether leading machine-learning algorithms (transformers) learn like brains. If transformers really do learn like brains, then they should develop the same structure as newborn brains when trained on the same prenatal data. We focused on the visual system. Neuroscientists have discovered three signatures that characterize the proto-organization of newborn visual systems: (1) edge sensitivity in early layers (Hubel & Wiesel, 1962; 1963; Priebe, 2016), (2) shape sensitivity in later layers (Brincat & Connor, 2004; Kourtzi & Kanwisher, 2001), and (3) hierarchical & retinotopic architectures (i.e., gradually increasing receptive field sizes across layers) (Arcaro & Livingstone, 2017; Ellis et al., 2021). These signatures are present at birth and constant across development (Arcaro & Livingstone, 2021). However, there are deep disagreements about how visual systems get this structure (Fig. 1). Do genes instruct the organization of visual systems (*Instructional Theories*)? Or do visual systems develop their organization as they fit to the organism's prenatal world (*Selectional Theories*)? We explored whether transformers—which initially lack all three signatures—spontaneously develop the same proto-organization as newborn visual systems when trained on retinal waves (Fig. 2).

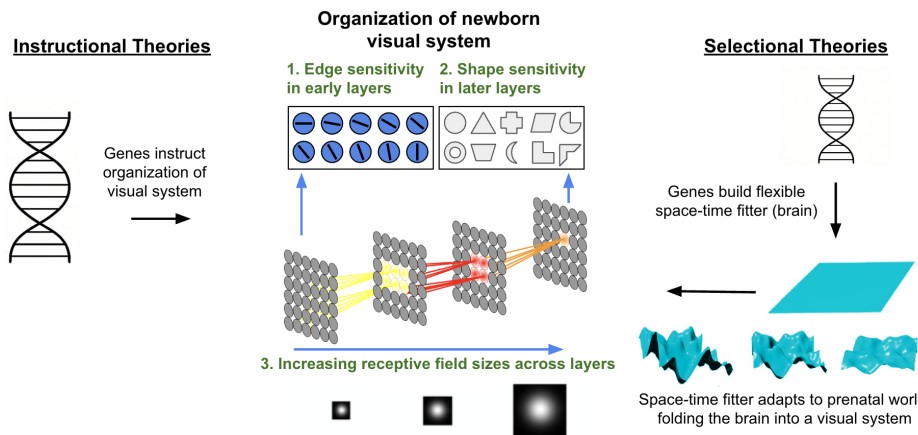

Figure 1: How do visual systems get their structure? Neuroscientists discovered that newborn visual systems are highly structured, including (1) edge sensitivity in early layers, (2) shape sensitivity in later layers, and (3) increasing receptive field sizes across layers. Instructional theories propose that genes instruct the organization of newborn visual systems. Selectional theories propose that genes produce a flexible space-time fitter (brain), which adapts to the continuous flow of prenatal sensory input. As brains adapt, they develop the structure of newborn visual systems.

Brains learn through unsupervised temporal learning (DiCarlo et al., 2012; Feldman & Tremoulet, 2006; Földiák, 1991; Rolls, 2012; Stone, 1996; Wiskott & Sejnowski, 2002), so we used space-time fitting transformers that also learn through unsupervised temporal learning (Fig. 2a). During training, the model pushes together embeddings of images in the same temporal window (300 ms), while pushing apart embeddings of non-temporally-adjacent images. This learning objective mimics unsupervised temporal learning in mature primates (Cox et al., 2005; Li & DiCarlo, 2008; Meyer & Olson, 2011; Wallis et al., 2009) and newborn animals (Matteucci & Zoccolan, 2020; Wood, 2016; Wood et al., 2016; Wood & Wood, 2018; 2016; 2021).

When transformers adapt to retinal waves (Fig. 2b), the models spontaneously develop the same structure as newborn visual systems (Fig. 2c). Transformers develop (1) edge sensitivity in early layers, (2) shape sensitivity in later layers, and (3) larger receptive fields across layers, mimicking the proto-organization of newborn visual systems. These results show that the organization of the visual system can be largely explained in terms of general space-time fitting principles.

## 1.1 RELATED WORK

**Training models on retinal waves.** Prior studies have trained models on retinal waves. Albert et al. (2008) found that independent component analysis algorithms learn orientation selective codes (resembling those in primary visual cortex) when trained on simulated retinal waves. Cappell et al. (2024) and Ligeralde et al. (2024) extended this approach to neural network models, showing that CNNs trained on retinal waves develop V1-like features, akin to the edge representations found in primary visual cortex.

While informative, these prior studies (1) did not use spatiotemporal retinal waves (the models were trained on static images), (2) did not use biologically inspired learning objectives (the models did not use unsupervised temporal learning), and (3) did not analyze how retinal wave training impacts the large-scale structure of the visual system (prior studies focused on primary visual cortex). Prior studies also used CNNs. CNNs have hardcoded spatial priors, including local connectivity, translation invariance, pooling layers, and spatial hierarchies. As such, even untrained CNNs have oriented edge representations and hierarchical & retinotopic architectures (Ulyanov et al., 2018). CNNs are hardcoded to mimic adult visual systems, so they cannot be used to explore how visual systems develop their structure in the first place (since the organization is hardcoded into the model).

Unlike CNNs, transformers are generic learners without hardcoded domain-specific priors. In their untrained state, transformers lack all three signatures of newborn visual systems. Transformers

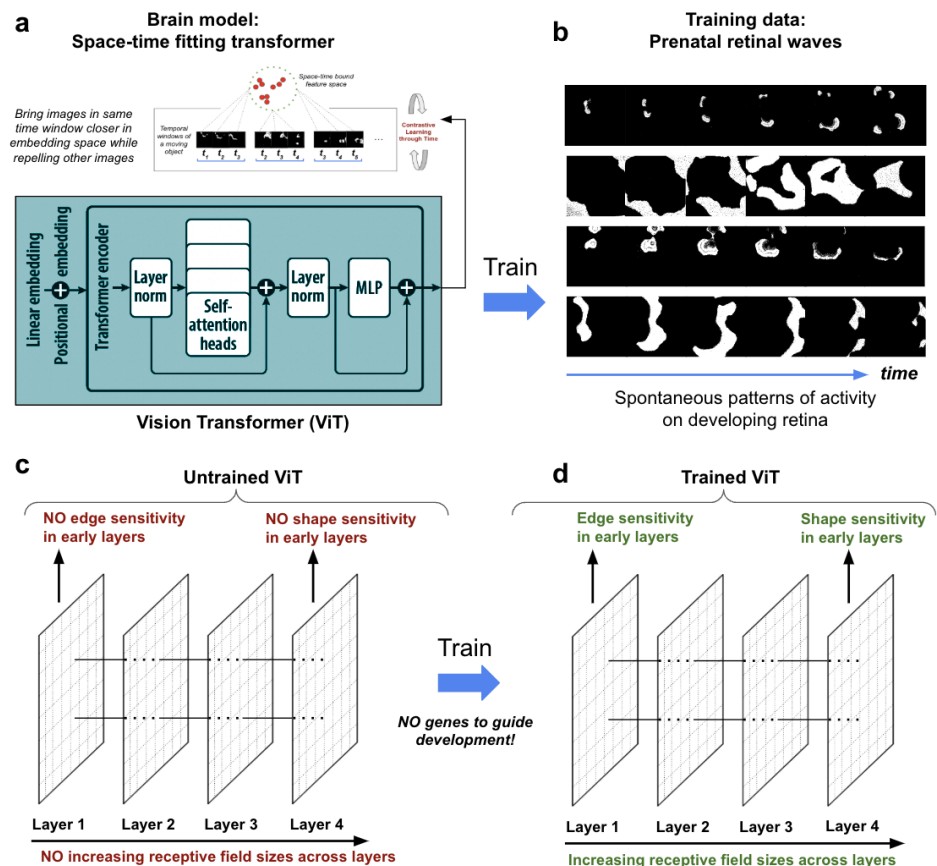

Figure 2: Testing Selectional Theories of visual development. If adaptation to space-time data distributions in prenatal worlds is sufficient to produce structured visual systems, then space-time fitting models should develop the same organization as newborn visual systems when trained in prenatal worlds. We tested this prediction by (a) selecting a space-time fitting transformer model, then (b) training the model on simulated prenatal experiences (retinal waves). (c) In their untrained state, the models were not organized like newborn visual systems: they lacked edge sensitivity in early layers, shape sensitivity in later layers, and increasing receptive field sizes across layers. (d) After fitting to retinal waves, the models spontaneously developed all three signatures of newborn visual systems. Space-time fitting transformers develop the same structure as newborn visual systems when exposed to the same prenatal data as newborns.

can thus reveal whether large-scale structure is learnable from experience, without hardcoded domain-specific priors to guide development. We tested whether transformers that learn through space-time fitting—adapting to the flow of experience from prenatal retinal waves—spontaneously develop edge sensitivity, shape sensitivity, and hierarchical & retinotopic architectures.

**Learning convolutional architectures.** Other studies have explored whether generic learning models can develop a retinotopic and hierarchical architecture. Ingrosso & Goldt (2022) found that when fully-connected neural networks are trained on data with non-Gaussian, higher-order local structure, the models develop the localized, space-tiling receptive fields that characterize CNNs. Raghavan & Thomson (2019) performed simulations showing that retinotopic networks can be grown from a single cell using two ingredients present during prenatal development: retinal waves and spike-timing-dependent plasticity. We extend these findings by training transformers on retinal waves, exploring whether the proto-organization of newborn visual systems is an emergent product of generic space-time fitting systems adapting to prenatal experience.

**How do brains get their structure?** Finally, our study tackles the longstanding question of how brains get their structure (Fig. 1). With 86 billion neurons and trillions of connections, human

brains are the most complex structures in the known universe. But where does this structure come from? How does a mass of brain tissue develop into a coordinated, specialized organ capable of perception, cognition, and action? Nativist (*instructional*) theories propose that brain structure is innate, encoded in the genome (Koffka, 1935; L'Abbate & Ratto, 1998; Carey, 2009; Spelke, 2022). Empiricist (*selectional*) theories propose that brain structure is the product of experience and learning (Wood et al., 2024; Thelen & Smith, 2007; Hasson et al., 2020). We tackle this debate by testing whether the complex structure of newborn visual systems can develop entirely from generic space-time fitters adapting to prenatal experience, without genes to instruct development in any way.

## 2 METHODS

### 2.1 MODELS

We used a Vision Transformer with Contrastive Learning through Time (ViT-CoT). The model has a ViT architecture and self-supervised contrastive learning through time learning objective that pushes together embeddings of images that appear in the same temporal window (three images, 300 ms), while separating embeddings of images that do not co-occur (Pandey et al., 2023). ViT-CoT is a generic space-time fitting model that starts with no hardcoded domain-specific priors (untrained state), then gradually adapts its representational space to fit its visual diet. We used three architecture sizes, with four, five, or six attention heads and layers. For example, the ViT-CoT (4H) architecture had four attention heads and four layers. To avoid hardcoding spatial priors into the transformers, no convolutional layers were used to create image patches. We used 64×64 resolution images and a patch size of 8×8. The models were optimized using the Adam optimizer with a constant learning rate of 0.0001, over 100 training epochs, with a batch size of 128. We maintained the original sequence of frames across all epochs, without shuffling.

### 2.2 TRAINING DATA

*Retinal Wave Simulation.* Retinal waves are spontaneous bursts of activity in retinal amacrine cells that propagate in a wave-like fashion during prenatal development (Fig. 2b). Researchers have developed several retinal wave simulators (Cessac et al., 2017; Godfrey & Swindale, 2007). We used the Retinal Wave Simulator (RWS) developed by Godfrey & Swindale (2007). RWS is publicly available and provides biologically accurate hyperparameter toggles for wave frequency, refractory period, spike rate, excitation time, and other variables. RWS also includes hyperparameter value presets for a range of species (e.g., chicks, ferrets, mice, rabbits, turtles), derived from neurophysiological retinal wave recordings. We used the hyperparameter values for chicks and generated retinal wave videos containing 160,000 images. The models were trained on different numbers of retinal waves to measure the impact of learning on the resulting structure of the visual system. After training, the models were frozen for the analyses described below.

### 2.3 MEASURING ORGANIZATIONAL STRUCTURE

*Edge Sensitivity.* We tested whether the models developed edge sensitivity in early layers, consistent with observations that early layers of visual systems (e.g., primary visual cortex) have oriented edge representations (Fig. 3). We generated a stimuli set containing ten line orientations ranging from 0° to 90° (Fig. 3a). As a baseline, we created a hardcoded orientation-tuned representational dissimilarity matrix (RDM) (Kriegeskorte et al., 2008) (Fig. 3c). In this hardcoded RDM, similarity between orientations decreased as a function of orientation distance. The RDM was a symmetric matrix with a diagonal of ones (indicating perfect self-correlation) and geometric progression with a decay factor of 0.5 when moving one step away (left or right) from the diagonal. The decay rate formalizes the idea that orientation tuning should produce increasingly dissimilar representations as line orientations diverge. To measure edge sensitivity, we computed the correlation between each model layer's RDM and the hardcoded orientation-tuned RDM. When computing correlations, we removed the diagonals from the RDMs to avoid artificially inflating the correlation (since the diagonals measure similarity between a representation and itself, which is not informative).

*Shape Sensitivity.* We tested whether the models developed shape sensitivity in later layers, consistent with observations that later layers of visual systems (e.g., inferior temporal cortex) have shape

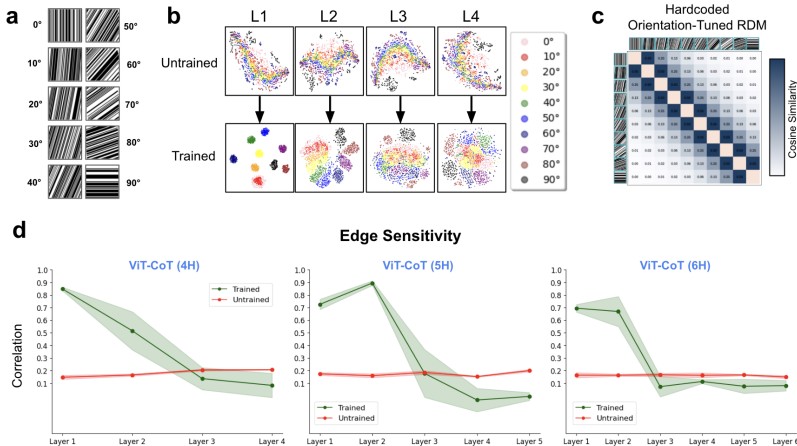

Figure 3: Edge development. (a) We measured how models organize lines with different orientations, separated by 10° increments. (b) t-SNE visualizations show that untrained models do not have structured feature spaces that organize images as a function of line orientation, whereas trained models develop structured feature spaces. Early layers (L) have particularly structured edge representations. (c) As a baseline, we created a hardcoded orientation-tuned RDM where similarity between orientations decreased as a function of distance. (d) Correlations between untrained (*red lines*) and trained (*green lines*) RDMs for each layer of the model and the orientation-tuned RDM. After training, early layers of the models developed sensitivity to oriented edges.

representations (Fig. 4). Specifically, we measured whether each layer prioritizes object contours over color features when grouping novel objects. We generated two stimuli sets, each containing 16 objects, composed of all combinations of four shapes and four colors (Fig. 4a). In Stimuli Set 1, we used simple shapes and colors to approximate the object stimuli used to study shape perception in infants. In Stimuli Set 2, we used realistic objects to test whether our results generalize beyond simple geometric shapes.

If a layer has a color-based representational space, then objects of the same color (but different shape) will group together (Fig. 4b). Conversely, if a layer has a shape-based representational space, then objects of the same shape (but different color) will group together (Fig. 4c). We analyzed the representational spaces of the model layers using RDMs. The RDMs show the distances between representations of image pairs in a layer's embedding space. With RDMs, we can visualize whether a layer builds more similar representations of objects that match in color (Fig. 4b) versus shape (Fig. 4c). To quantify whether models were color-based versus shape-based, we computed the average representational similarity between objects that matched in color (color score, Fig. 4b) versus shape (shape score, Fig. 4c).

*Receptive field sizes across layers.* We tested whether the models developed a CNN-like (hierarchical and retinotopic) architecture, consistent with reports that newborn visual systems have progressively larger receptive field sizes across areas (Arcaro & Livingstone, 2021) (Fig. 5). To assess this, we adopted the method used by Huang et al. (2024) for quantifying receptive fields in Vision Transformer models. This gradient-based method estimates receptive fields by measuring which input regions most affect the model's output, capturing the spatial sensitivity at each layer. Specifically, we passed test images of a moving object (Fig. S3) through our frozen ViT-CoT model and recorded the logits at each intermediate layer during the forward pass. To estimate the receptive field, we selected a central region of units (neurons) in each layer and computed the gradients of their activations (logits) with respect to the input pixels via backpropagation during the backward pass. This process was repeated for each layer in the ViT-CoT model, and the resulting gradients were used to generate a map representing the receptive field of each layer. To quantify the receptive field, we measured the *receptive field size*, which reflects variance—the spatial extent of input influence on the selected units. We explain the algorithm in detail in Appendix (A.3.1).

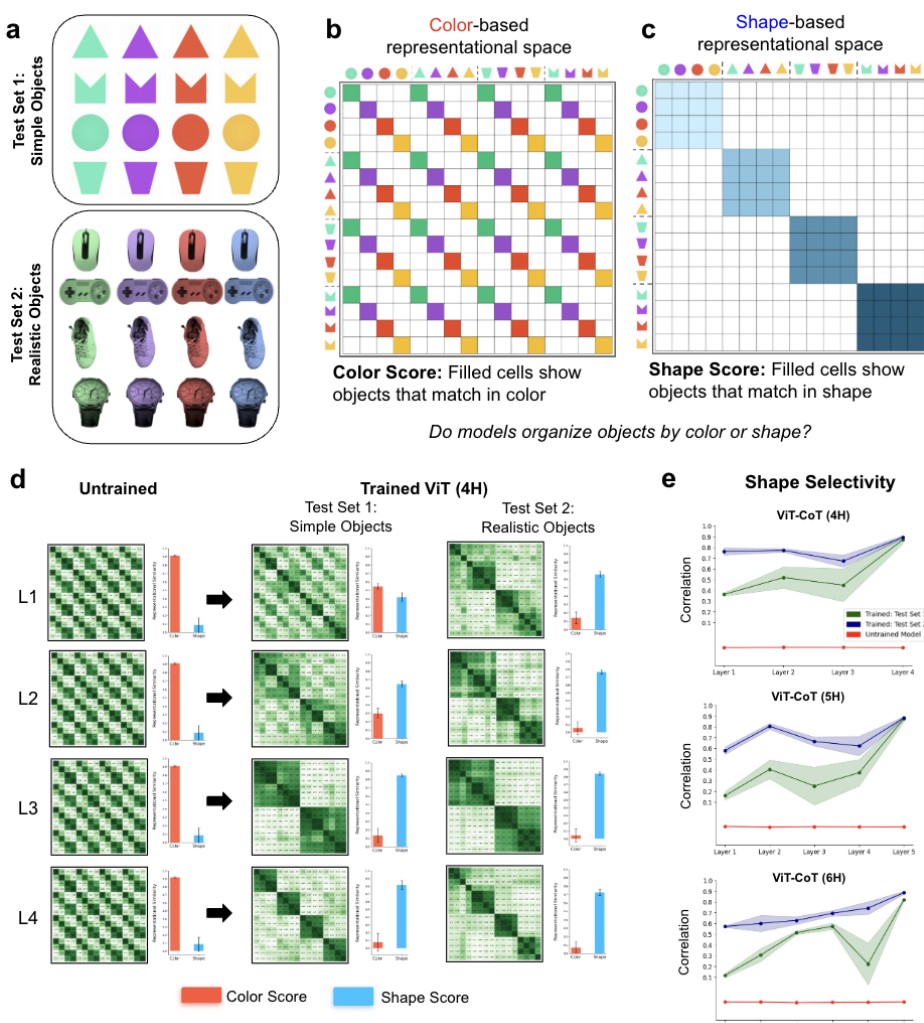

Figure 4: Shape development. (a) We tested whether models group novel objects by color versus shape, using two test sets (simple and realistic objects), each crossing four colors and four shapes. RDMs were used to evaluate whether trained fitting models developed color-based versus shape-based representational spaces. (b) Color scores were computed by averaging across RDM cells where objects matched in color (filled cells in Panel b). (c) Shape scores were computed by averaging across RDM cells where objects matched in shape (filled cells in Panel c). (d) Untrained models had color-based representational spaces, whereas trained models developed shape-based representational spaces, as demonstrated by the RDMs (*left*) and color/shape scores (*right*). (e) The correlations between untrained (*red lines*) and trained (test set 1: *green lines*; test set 2: *blue lines*) RDMs for each layer of the model and the shape-tuned RDM (panel c). After training, the models developed sensitivity to object shape.

## 3 RESULTS

**Edge sensitivity**. t-SNE visualizations (Fig. 3b) show that untrained models do not have structured feature spaces that organize images as a function of line orientation, whereas trained models develop structured feature spaces. Early layers have particularly structured edge representations. Fig. 3d shows the correlations between the model layer RDMs and the hardcoded orientation-tuned RDM. In their untrained state, transformers lacked oriented edge representations in all layers (Fig. 3d, red lines). This pattern was observed across the small (4H), medium (5H), and large (6H) architecture sizes. After being trained on retinal waves, the transformers developed robust

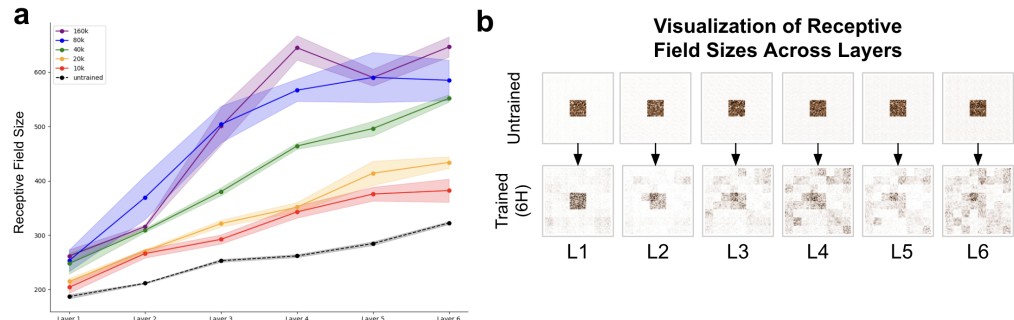

Figure 5: Development of larger receptive field (RF) sizes across layers. (a) We measured RF sizes across all layers of the models. In the trained models (*colored lines*), RF sizes increased across layers. Different colors indicate models trained on different numbers of retinal waves. In contrast, untrained models (*black line*) showed minimal increases in RF size across layers. (b) RF sizes were estimated using a gradient-based method that tracks sensitivity to input pixels. This method estimates RF sizes by identifying which input pixels influence the output of each layer. The gradient maps for untrained models *(top)* are similar across all layers, indicating a lack of hierarchical spatial organization and no effective increase in RF size. The gradient maps for trained models *(bottom)* become progressively broader across layers, indicating an increase in RF size and the emergence of hierarchical spatial organization.

oriented edge representations in early layers (Fig. 3d, *green lines*). This pattern was again observed across the small, medium, and large architecture sizes. To measure the impact of learning on the development of edge sensitivity, we trained models on different numbers of retinal waves, ranging from 10k to 160k images (Fig. S1). When models were trained on larger numbers of retinal waves, they developed more structured edge representations in early layers. Space-time adaptation to retinal waves causes transformers to develop orientation selectivity in early layers, akin to the edge representations found in newborn visual systems.

**Shape sensitivity**. RDMs showed that untrained models had color-based representational spaces, whereas trained models developed shape-based representational spaces (Fig. 4d). Fig. 4e shows the shape sensitivity scores across the model layers. In their untrained state, transformers lacked shape sensitivity in all layers (Fig. 4e, *red lines*). This pattern was observed across the small, medium, and large architecture sizes. After being trained on retinal waves, the transformers developed shape sensitivity in later layers (Fig. 4e, *green lines*). This pattern was again observed across the small, medium, and large architecture sizes. To measure the impact of learning on the development of shape sensitivity, we trained models on different numbers of retinal waves, ranging from 10k to 160k images (Fig. S2). When models were trained on larger numbers of retinal waves, they developed more robust shape sensitivity. Space-time adaptation to retinal waves causes transformers to develop shape-based representational spaces; these spaces are more sensitive to the contours, rather than colors, of objects. Like biological visual systems, shape sensitive regions tend to be more robust in later layers of the system.

**Larger receptive fields across layers**. In their untrained state, the models' receptive field sizes did not increase substantially across layers (Fig. 5a, *dotted black line*). This pattern was observed across the small, medium, and large architecture sizes. After being trained on retinal waves, the models developed larger receptive fields across layers (Fig. 5a, *purple line*). This pattern was again observed across the small, medium, and large architecture sizes. To measure the impact of learning on the development of this organization, we trained models on different numbers of retinal waves (Fig. 5a, *colored lines*). When models were trained on larger numbers of retinal waves, they developed progressively larger receptive fields across layers. Fig. 5b visualizes the receptive fields of untrained and trained models across layers. Space-time adaptation to retinal waves causes transformers to develop larger receptive fields across layers, mimicking the architecture of newborn visual systems.

**Temporal scrambling** We hypothesize that adaptation to the temporal flow of prenatal experience determines the structure of visual systems. To test the importance of temporal information in developing a newborn-like visual system, we reran the experiments, but scrambled the order of the retinal

images (Fig. 6a). Rather than learning from retinal waves that moved smoothly over time, the models learned from retinal waves that moved non-smoothly. When trained in temporally non-smooth worlds, the models did not self-organize like newborn visual systems. The models failed to develop edge selectivity (Fig. 6b), shape selectivity (Fig. 6c), and larger receptive fields across layers (Fig. 6d). Thus, space-time fitting transformers self-organize into newborn-like visual systems when trained on temporally smooth visual experiences. Smoothness constraints on visual learning have been observed in newborn animals (Wood & Wood, 2018; Matteucci & Zoccolan, 2020), suggesting common learning constraints in brains and transformers.

## 4 DISCUSSION

Our results show that when transformers are trained in a prenatal visual world—adapting to the flow of sensory experiences produced by retinal waves—the models spontaneously develop the same structure as newborn visual systems. Transformers develop (1) edge sensitivity in early layers, (2) shape sensitivity in later layers, and (3) larger receptive fields across layers, mimicking the proto-organization of newborn visual systems. This developmental convergence in structure across brains

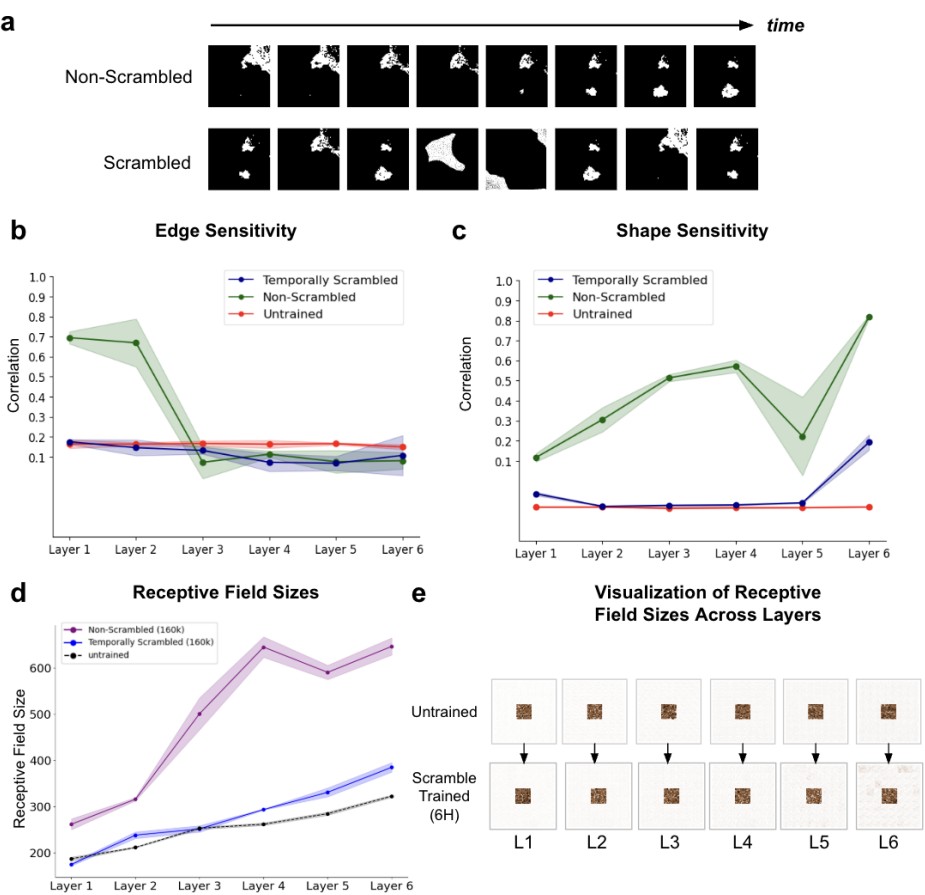

Figure 6: Temporal scrambling. (a) We trained models using either normal *(top)* or temporally scrambled *(bottom)* retinal waves. In the scrambled condition, both the temporal windows and the images within each window were randomly shuffled in each epoch to disrupt temporal continuity. Models trained on scrambled retinal waves self-organized to resemble untrained models, including (b) no edge sensitivity in early layers, (c) no shape sensitivity in later layers, and (d) minimal changes in receptive field sizes across layers. (e) The gradient maps for untrained models *(top)* and models trained on temporally scrambled retinal waves *(bottom)* were similar across layers, indicating a lack of hierarchical spatial organization

and transformers indicates that they learn in common ways. We argue that transformers can be used to model prenatal and postnatal brain development, opening new paths for science and engineering.

**Scientific Implications.** For science, we provide an image-computable explanation for the structure of the visual system. For centuries, scientists have debated why visual systems have the organization that they do. Some theories have emphasized the importance of genes for instructing the initial organization of visual systems, whereas others have emphasized the importance of learning and experience on the development of brain structure. But due to limitations in models and compute power, it was not previously possible to test how much structure could develop from experience alone, without genes to guide development. Using transformers, we show that the large-scale organization of newborn visual systems can emerge entirely from adaptation to prenatal experience. If space-time adaptation during prenatal development is sufficient to reproduce the structure of newborn visual systems, then it is unnecessary to postulate a central role for genes in guiding the organization of visual systems.

Looking forward, space-time fitting transformers can serve as baseline models for exploring whether genetic instruction is needed to develop brain-like organization. We can ask via simulation: If a visual system's organization depends entirely on prenatal adaptation—with no genes to guide development—what does the resulting organization look like? Our results show that it looks like a newborn visual system. Statistically, the organization of newborn visual systems is what we would expect if brains develop their structure from a flexible space-time fitting system adapting to prenatal experience.

**Engineering implications.** Transformers have revolutionized AI and penetrated deeply into the public's everyday technology use. How did this happen? Did engineers create a new "alien" form of intelligence that operates by different principles than biological intelligence? Or do transformers mimic key computational principles of brains, allowing transformers to approximate biological learning? Distinguishing between these possibilities is essential both for understanding the kind of intelligence that engineers have created and for communicating this technology to the public. Our results provide a new kind of evidence that transformers and brains learn in common ways: when given prenatal visual experiences, transformers develop newborn-like visual systems.

There is growing support for the conclusion that transformers learn like brains. For example, a transformer's computations approximate the internal mechanics and outputs of neuron-astrocyte networks in brains (Kozachkov et al., 2023), and waves of neural activity allow temporal context to be extracted from sequences of sensory inputs, approximating the self-attention computation of transformers (Muller et al., 2024). Powered by these core computations, transformers have excelled on vision tasks, while learning features that can be used to predict neural activation patterns in adult brains (Conwell et al., 2025). Transformers also learn internal representations similar to those found in brains, including grid cells, place cells, border cells, and direction cells (Whittington et al., 2021). Developmentally, transformers can learn to recognize objects when reared in the same impoverished environments as newborn chicks in controlled-rearing studies, including environments with a single object (Pandey et al., 2023). Transformers also show the same pattern of successes and failures as newborn chicks across visual learning tasks (Pandey et al., 2024). Overall, transformers appear to be accurate models for predicting the learning outcomes of brains.

**Limitations** Our study raises many new questions. First, we tested for three key signatures of newborn visual systems, but future studies could explore whether other characteristics of visual systems also develop spontaneously when space-time fitting transformers adapt to prenatal experience. Second, do other learning objectives produce similar organizational structures? Future studies could test whether transformers equipped with alternative unsupervised temporal learning objectives also self-organize like newborn visual systems. Third, what role do other prenatal experiences play in the development of brain structure? While we simulated prenatal visual experiences, future work could simulate prenatal input in other domains, including auditory, proprioceptive, and tactile experiences.

**Broader Impact** Our results suggest a new path for closing the learning gap between humans and machines. Humans have a long prenatal developmental phase, which provides rich experiences that shape the initial structure of the brain. Transformers, in contrast, do not have a prenatal developmental phase. We speculate that giving transformers a prenatal developmental phase will make them more brain-like, allowing models to develop the same proto-organization as brains before engaging in postnatal learning about the world.

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

## A    APPENDIX

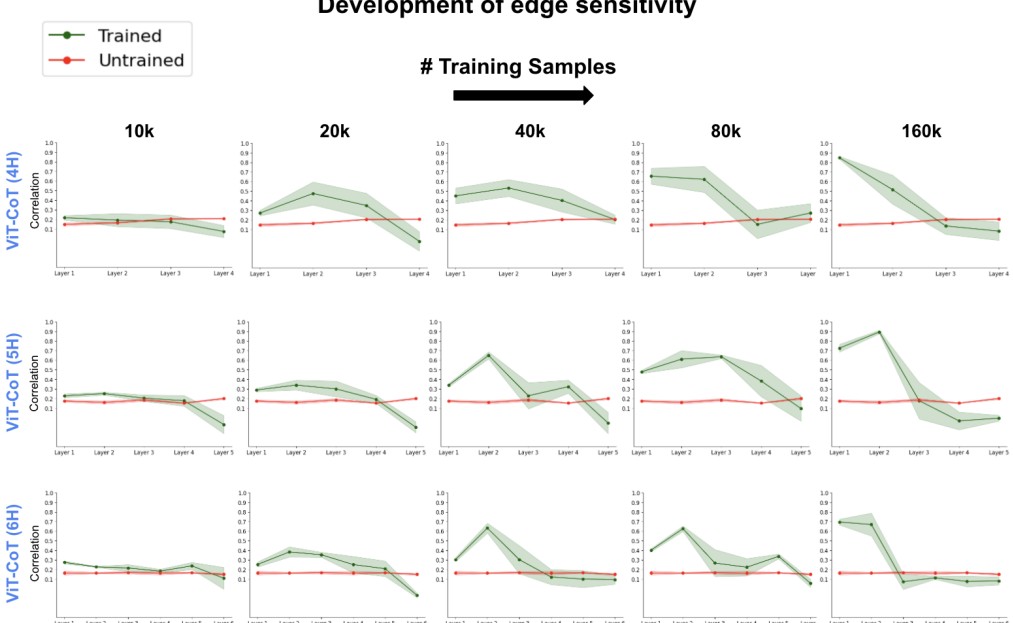

Figure S1: To measure the impact of learning on the development of edge sensitivity, we trained models on different numbers of retinal waves, ranging from 10k to 160k images. As models were trained on larger numbers of retinal waves, they developed more robust edge representations in early layers.

### A.1    TRAINING DETAILS

Table 1: Architectures and Hyperparameters for ViT-CoTs

| Model | Parameters (M) | Attention Heads | Layers | Learning Objective |
|---|---|---|---|---|
| ViT-CoT (4H) | 23.0 | 4 | 4 | Contrastive Learning Through Time |
| ViT-CoT (5H) | 29.5 | 5 | 5 | Contrastive Learning Through Time |
| ViT-CoT (6H) | 36.4 | 6 | 6 | Contrastive Learning Through Time |

Each vision transformer model was trained using three different seeds, each initialized with different random weights. To avoid hardcoding spatial priors into the models, we did not use any convolutional layers to create image patches. We used 64x64 resolution images with a patch size of 8x8 to train the models. The models were optimized using the Adam optimizer with a constant learning rate of 0.0001, over 100 training epochs, with a batch size of 128 frames. Our smallest model, ViT-CoT (4H), had 23.0 M trainable parameters, while our largest model, ViT-CoT (6H), had 36.4 M trainable parameters (Table 1). All models were trained on a single NVIDIA A10 GPU with 24 gigabyte of memory.

We trained the models in two conditions. In the *non-scrambled condition*, we did not shuffle the images and passed the images sequentially to the model during training to preserve the smooth temporal continuity of the retinal waves. In the *scrambled condition*, we shuffled both the temporal windows and the frames within the windows after every training epoch, breaking the temporal smoothness of the retinal waves.

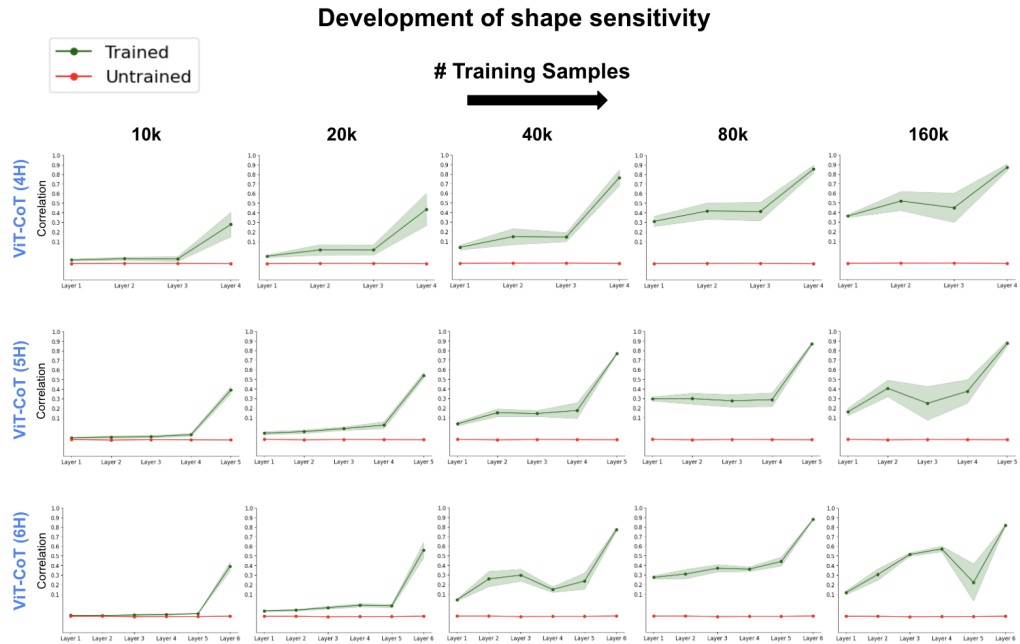

Figure S2: To measure the impact of learning on the development of shape sensitivity, we trained models on different numbers of retinal waves, ranging from 10k to 160k images. As models were trained on larger numbers of retinal waves, they developed more robust shape sensitivity in later layers. Space-time fitting to retinal waves causes transformers to develop shape-based representational spaces; these spaces are more sensitive to the contours, rather than colors, of objects.

## A.2 STIMULI GENERATION

### A.2.1 LINE ORIENTATIONS STIMULI

To generate lines of different orientations, we first created a 1D NumPy array and randomly populated it with 0s and 1s. We then reshaped it into a single-row image and plotted it using a binary color map, where 0s represented white and 1s represented black. When displayed with a stretched horizontal aspect ratio, this arrangement produced a series of vertical black-and-white bars—effectively forming line patterns. Finally, we rotated the images and created ten different orientation sets, with the sets separated by 10° increments.

### A.2.2 RECEPTIVE FIELD MAPPING STIMULI

The receptive field sizes were measured using a mini batch of 29 frames collected from the shape shown in Fig. S3. We projected the object on a virtual monitor in a simulation platform (Unity) and collected 29 frames while the object smoothly rotated from side to side on a white background (Fig. S3).

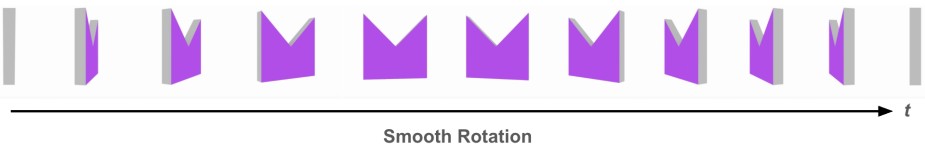

Smooth Rotation

Figure S3: To visualize gradient maps and quantify receptive field sizes across layers, we collected 29 images from this shape as it rotated from side to side on a white background. A mini-batch of 29 frames was created to visualize the receptive fields.

## A.3 ALGORITHMS

### A.3.1 RECEPTIVE FIELD ANALYSIS

We performed receptive field analysis on the models to check for hierarchal and retinotopic structures in the network. Below we explain the algorithm for visualizing and quantifying the receptive field of each layer of the model:

Let

$$f_q : \mathbb{R}^{B \times C \times H \times W} \to \mathbb{R}^{B \times D} \tag{1}$$

be the trained and frozen Vision Transformer encoder with parameters $q$, where:

- $B$ is the batch size,
- $C$ is the number of channels,
- $H$ is the height of the input image,
- $W$ is the width of the input image, and
- $D$ is the feature dimension.

Let

$$x \in \mathbb{R}^{B \times C \times H \times W} \tag{2}$$

be a batch of test samples. Instead of taking the final output $f_q(x)$, we extract intermediate representations from each transformer layer:

$$f_q^{(l)}(x) \in \mathbb{R}^{B \times (1+T) \times d}, \quad \text{for each layer } l = 1, \ldots, L \tag{3}$$

where:

- $(1 + T)$ is the number of patch tokens (including the class token),
- $d$ is the embedding dimension,
- $f_q^{(l)}(x)$ is the output (logits) from the $l^{\text{th}}$ layer.

We remove the class token and reshape the output tensor into a 2D spatial grid (assuming $T = P^2$):

$$z^{(l)} = f_q^{(l)}(x)[:, 1 :, :] \in \mathbb{R}^{B \times T \times d} \tag{4}$$

$$z^{(l)} \to \tilde{z}^{(l)} \in \mathbb{R}^{B \times P \times P \times d} \tag{5}$$

From this 2D spatial grid, we extract the central region of size $p \times p$ (e.g., $p = 3$ or $p = 4$) with $p < P$:

Let

$$s = \left\lfloor \frac{P - p}{2} \right\rfloor, \quad \text{then}$$

$$z_{center}^{(l)} = \tilde{z}^{(l)}[:, s : s + p, s : s + p, :] \in \mathbb{R}^{B \times p \times p \times d} \tag{6}$$

The tensor is then reshaped to flatten the spatial dimensions as follows:

$$\tilde{z}^{(l)} = reshape(z_{center}^{(l)}) \in \mathbb{R}^{B \times (p^2) \times d} \tag{7}$$

Next, we calculate the scaler activation by taking the mean across spatial patches and then the embedding dimension to backpropagate:

$$a^{(l)} = \frac{1}{p^2} \sum_{i=1}^{p^2} \hat{z}^{(l)}[:, i, :] \in \mathbb{R}^{B \times d} \tag{8}$$

$$s^{(l)} = \frac{1}{d} \sum_{j=1}^{d} a^{(l)}[:,j] \in \mathbb{R}^B \tag{9}$$

Finally, we backpropagate to input. Let gradients of $s^l$ w.r.t input be:

$$g^{(l)} = \left| \frac{\partial s^{(l)}}{\partial x} \right| \in \mathbb{R}^{B \times C \times H \times W} \tag{10}$$

The output is then normalized before visualizing:

$$\tilde{g}^{(l)} = \frac{g^{(l)} - \min(g^{(l)})}{\max(g^{(l)}) - \min(g^{(l)})} \in [0,1]^{B \times C \times H \times W} \tag{11}$$

We show the gradient map plot in eq (11) in Fig. 5b and Fig. 6e in the main text.

To quantify the receptive field from the gradient map in (11), we measure the receptive field size by calculating the spatial variance. We first normalize the gradient map from eq (11) so that it's total sum is 1, representing a probability distribution over spatial locations:

$$\sum_{i=1}^{H} \sum_{j=1}^{W} \tilde{g}_{i,j} = 1 \tag{12}$$

$$x_{\text{mean}} = \sum_{i=1}^{H} \sum_{j=1}^{W} j \cdot \tilde{g}_{i,j}, \quad y_{\text{mean}} = \sum_{i=1}^{H} \sum_{j=1}^{W} i \cdot \tilde{g}_{i,j} \tag{13}$$

where, each position $(i,j)$ is weighted by how strong the gradient was there (from $\tilde{g}$), indicating where the model 'looks' the most.

Finally, we calculate the *spatial variance*, a measure of how spread out the information is:

$$\text{Var}^{(l)} = \sum_{i=1}^{H} \sum_{j=1}^{W} \left[ (j - x_{\text{mean}})^2 + (i - y_{\text{mean}})^2 \right] \cdot \tilde{g}_{i,j} \tag{14}$$

### A.3.2 RDM CONSTRUCTION

We evaluated the models using Representational Dissimilarity Matrices (RDMs), an unsupervised analysis method. To generate the RDMs, we first froze the model weights and passed individual test samples through the network to extract their features. These features were then normalized by subtracting the mean feature vector from each sample's feature vector. Finally, we computed pairwise cosine similarities between the normalized features and visualized the resulting similarity scores as RDMs.

For intermediate layers that include an additional dimension representing patch tokens along with a class token, we first removed the class token and then applied average pooling across the patch token dimension. The resulting feature vectors were then processed using the same steps described above.

### A.4 DATA AND CODE AVAILABILITY

The code and data needed to reproduce these findings will be provided in a GitHub repository upon publication.

