# OpenReview forum: "Transformers self-organize like newborn visual systems when trained in prenatal worlds"
_ICLR.cc/2026/Conference — ICLR 2026 Conference Withdrawn Submission_

### Official Review · Reviewer_czJx · 2025-10-29

**Soundness:** 2
**Presentation:** 3
**Contribution:** 2
**Rating:** 2
**Confidence:** 3

**Summary:**

This paper examines a specific class of flexible space-time fitters, i.e., transformers, by training them on retinal wave-like visual stimuli to recapitulate edge specificity in early model layers, shape specificity in later model layers, and a local-to-global effective receptive field hierarchy across the model.

**Strengths:**

Understanding the functio-structural relevance of retinal waves is an important question as a lot of prior research has theorized their involvement in shaping the functional organization of the cortex in-utero. If one were to build a biophysically-plausible model of the visual system to analyze developmental similarities, then an investigation into some form of pre-eye opening activity-dependent organization of the model units seems necessary. The paper does a good job of presenting the motivation, and the paper is generally easy to follow. The methods used also seem reasonable.

**Weaknesses:**

I am a bit confused about the term "self-organization" as used throughout the paper. When I think about self-organization or functional organization or organization more simply, I dissociate it from representational similarity. The methods used in the paper, such as for edge and shape selectivity, either simply compare representational similarities between the model layers and that hardcoded for orientation tuning, or discriminate between color- and shape-based preference. How do the RSMs from different model layers compare to those from humans? Prior works have already compared RSMs between models and adult brains, so using some of that data here might be helpful towards the claim. But thinking outside of just representational similarity, a leading theory behind why brains functionally organize is due to wiring economy constraints, such as that explored by Margalit et al. (2024). If the claim being made in the paper is that transformers "mimick the proto-organization of newborn visual systems" (ll. 392-393), then the methods as they have been used in the paper do not fully support that. How should I think about organization without considering the model as topographic at some level?

Additionally, I think the paper is missing a control. What happens if you train transformers on other kinds of stimuli, such as drifting sine gratings or natural videos, to tease out the unique contributions of the specific prenatal visual worlds (retinal waves) that infants are exposed to. Does pre-training, in any form irrespective of the visual world, yield the same forms of preferences in the model? If so, what roles do retinal waves play?

Finally, some of the figures such as 4(d) bar plots are really low-resolution and I cannot interpret them carefully even after significantly zooming in.

**Questions:**

Please see the weaknesses section above!

---

### Official Review · Reviewer_VdaR · 2025-10-30

**Soundness:** 3
**Presentation:** 3
**Contribution:** 2
**Rating:** 6
**Confidence:** 3

**Summary:**

The authors propose to train a Transformer on synthetically generated retinal wave data to simulate visual learning in prenatal phase. The resulting model is used as a model of the biological visual systems to probe various characteristics of proto-organization of visual systems like edge sensitivity, shape sensitivity, and increased receptive field size. The authors find that network trained on retinal wave data does in fact develop edge sensitivity in earlier layers, shape sensitivity to later layers, and increased receptive field size. The authors also train on temporal shuffled retinal wave data, and the network fails to learn lots of important structure of visual systems in prenatal phase. This indicate the necessity of the temporal statistics of retinal wave in facilitating basic visual system development.

**Strengths:**

- The paper is well-written and easy to follow.
- The neuroscientific motivation is strong. In the absence of better non-invasive device of recording the brain activities of prenatal babies, probing a computational model of the brain is the obvious way to go.

**Weaknesses:**

- Line 098: “did not use spatiotemporal retinal waves”. This is not quite true. Ligeralde et al. (2024) used spatiotemporal retinal waves data and also retinal waves data collected from neurophysical experiments.
- I have some concerns about the experiment over temporally shuffled retinal wave data. Essentially, if you destroy the temporally smoothly varying structure, then there is no hope of learning the conventional image-to-image similarity metric using any temporally contrastive methods. In fact, you are learning a distribution that’s already different from the natural video distribution. By pulling together representations in a small window of temporally shuffled data, you fail to isolate the effect of spatial statistics in facilitating the formations of various characteristics of prenatal visual systems. So it seems like to me that if you want to carefully isolate out the effect of spatial statistics in the absence of temporal statistics, you should use a reconstruction-based objective (either in pixel-space or latent space) over the retinal wave data. Otherwise, it’s hard to entangle whether the failure of learning structured receptive fields is due to absence of both spatial and temporal structure or just the temporal statistics.

**Questions:**

N/A

---

### Official Review · Reviewer_1rmk · 2025-10-31

**Soundness:** 2
**Presentation:** 2
**Contribution:** 2
**Rating:** 2
**Confidence:** 5

**Summary:**

This paper examines whether transformers can self-organize like newborn visual systems when trained on biologically inspired “prenatal” data. Specifically, the authors train Vision Transformers (ViT-CoT) on simulated retinal wave inputs using a temporally contrastive learning objective. They report that the trained models spontaneously develop three hallmarks of early visual organization: (1) edge sensitivity in early layers (2) shape sensitivity in later layers and (3) progressively larger receptive fields across depth. Control experiments using temporally scrambled inputs fail to produce these properties, leading the authors to conclude that temporal continuity in prenatal sensory experience is sufficient for the emergence of newborn-like visual structure.

While the framing is interesting, the main claims of the study seem overstated. It remains unclear what unique scientific or methodological insights arise specifically from using transformers instead of CNNs, beyond substituting architectures and adopting an existing temporal contrastive objective. The lack of downstream or out-of-domain evaluations makes it difficult to assess whether pretraining on retinal waves produces useful or generalizable representations. Training for 100 epochs (!) on a small, repetitive dataset diverges sharply from real prenatal conditions, and the model’s spatial resolution and foveal focus limit biological plausibility. The study seems to be an illustrative demonstration than quantitative progress toward understanding how brains self-organize.

**Strengths:**

1. he paper is well structured, and easy to follow from motivation through methods and results. The narrative is coherent, and the figures are well integrated with the text.
2. The plots and visualizations are clean, intuitive, and easy to interpret, which makes the main findings immediately accessible even to readers outside the specific subfield.
3. The idea itself is conceptually compelling and has the potential to inspire new directions at the intersection of neuroscience and machine learning.

**Weaknesses:**

1. The work closely mirrors the approach of [Ligeralde et al. (2024)], differing mainly in the substitution of CNNs with ViTs and the adoption of the ViT-CoT contrastive temporal loss from [Pandey et al.]. Several prior studies have already trained models on simulated retinal wave data, demonstrating the emergence of V1-like receptive fields. As a result, the scientific contribution here feels incremental (architecture substitution than fundamentally new idea or learning principle, or even a strong test)

2. The paper does not meaningfully compare the current results against other (prior) models trained on retinal activity (like Ligeralde et al., or ReWaRD). Is the claim that those models don't learn V1-like receptive fields? Not including the existing body of work within the evaluations of the current paper makes it unclear what conceptual advance is being claimed.

3. The authors claim that “if transformers really do learn like brains, they should develop the same structure as newborn brains when trained on the same prenatal data.” But they end up training , training for *100 epochs* (!) seems biologically implausible. This seems contradictory to the core premise of the paper. A model of development should consider the developmental training trajectories as well!

5. The experiments use very low-resolution inputs, which raises concerns about whether the reported findings would hold at more realistic scales (e.g., 224×224 images). Alternative architectures such as [LightViT](https://arxiv.org/pdf/2207.05557) could be used because a) they don’t have convolutional layers and b) they support large image resolutions.

6. The receptive field experiments are restricted to the foveal region. However, receptive field size is known to vary with eccentricity (see [Freeman & Simoncelli, 2011](https://www.nature.com/articles/nn.2889), Fig. 1). Extending this analysis across eccentricities would provide a more comprehensive and biologically grounded evaluation.

7. By default, the ViT-CoT architecture (as implemented in the [`vit_pytorch`](https://github.com/buildingamind/ViT-CoT/blob/097f83ed70814793f7b32a1beddeb3a4cbbc4625/requirements.txt#L19C1-L19C12) library used by [used by Pandey et al.](https://github.com/buildingamind/ViT-CoT/blob/097f83ed70814793f7b32a1beddeb3a4cbbc4625/models/vit_contrastive.py#L82), which the authors follow) takes the `[CLS]` token as the model’s output embedding by default (as seen [here](https://github.com/lucidrains/vit-pytorch/blob/cbf6723063df2aa89526f9482b1c9a64feef9cb0/vit_pytorch/vit.py#L83)). If this token is excluded when computing receptive fields, the analysis may no longer reflect the representations actually used for downstream processing. This discrepancy (using one token for functional output and another set for representational analysis) raises concerns about the validity and interpretability of the receptive field results.

**Questions:**

1. What does using ViTs and the ViT-CoT loss reveal beyond Ligeralde et al. (2024)? Is there any new scientific insight here, or just an architectural substitution? Do the authors think Transformers are more biologically realistic than CNNs? Please clarify the core claims better.

2. Why are prior models trained on retinal activity (Ligeralde et al., ReWaRD) not compared during evaluation? Do those models fail to show the same effects claimed in the study?

3. How is training for 100 epochs on a small, repeating dataset biologically plausible? Would a single-pass or variable-input setup yield the same outcomes?

4. Do the reported trends (edge → shape → hierarchy) hold at higher image resolutions (e.g., 224×224)? Why not test this with transformer variants like LightViT that can handle larger inputs?

5. Since receptive field size varies with eccentricity, have the authors tested whether similar scaling trends emerge beyond the foveal region?

6. Why does the receptive field analysis use patch tokens instead of the [CLS] token that defines the model’s output? Does this mismatch affect the validity of the results?

---

### Official Review · Reviewer_Z8zP · 2025-11-01

**Soundness:** 4
**Presentation:** 3
**Contribution:** 3
**Rating:** 8
**Confidence:** 4

**Summary:**

They train a transformer on simulated retinal waves. After training, the transformers develop common organizational patterns with the visual system, namely (1) edge detectors in early layers; (2) shape sensitivity (vs color) in later layers; (3) receptive field size increasing with layer depth.

**Strengths:**

- Original study.
- Carefully done analyses which support the claims of the study.
- Clear writing.
- Results with some significance to the field of neuroscience.
- Very good literature review.

**Weaknesses:**

- The model is trained with backprop, which limits the biological plausibility of the model. Alternatives have been proposed elsewhere, see e.g., https://openreview.net/forum?id=lQBsLfAWhj
- In the methods, I have not seen the description of the unsupervised temporal learning method. Please add.

**Questions:**

- Why was the study not including ConvNets as well?
- Consider citing https://arxiv.org/abs/2103.13023, with analyses and results which bear some resemblance to your work.
- fig2a: text too small

---

> ### Comment · Reviewer_Z8zP · 2025-11-25
>
> EDITED: In the absence of a convincing answer of the authors to the other reviews, I am not sure that the paper should be accepted, and my score no longer reflects my opinion of the paper.

---

### Note · Authors · 2025-11-29

**Comment:**

We thank all reviewers for their thoughtful and constructive feedback. We will carefully consider the comments and use them to strengthen our work. At this time, we have decided to withdraw the paper, as we are unable to incorporate the requested revisions within the available timeframe.

**Withdrawal Confirmation:**

I have read and agree with the venue's withdrawal policy on behalf of myself and my co-authors.